# Screening of Developmental Difficulties during the Transition to Primary School

**DOI:** 10.3390/ijerph18083958

**Published:** 2021-04-09

**Authors:** Carolina González, Ramón D. Castillo, José Patricio Franzani, Cristian Martinich

**Affiliations:** 1Parenting and Family Support Centre, School of Psychology, The University of Queensland, Brisbane, QLD 4072, Australia; c.gonzalez@uq.edu.au; 2Centro de Investigación en Ciencias Cognitivas, Facultad de Psicología, Universidad de Talca, Avenida Lircay s/n, Talca 3460000, Chile; 3Corporación para el Desarrollo del Aprendizaje (CDA), Santiago 8370167, Chile; jpfranzanic@gmail.com (J.P.F.); cdamartinich@yahoo.es (C.M.); 4Departamento de Humanidades y Educación, Universidad Tecnológica de Chile Inacap, Santiago 8320000, Chile

**Keywords:** gender differences, kindergartners, first graders, transition to primary school, Spanish version FTF questionnaire

## Abstract

The five-to-fifteen (FTF) questionnaire is a screening tool completed by parents that is able to distinguish developmental disorders in children aged 5 to 15 years old. The current study aimed to characterize the developmental difficulties by gender and school age (kindergarten and first grade) of children in their transition to primary school, using the Spanish-language version of the FTF questionnaire. The participants were 541 parents of typically developed children from kindergarten and first grade in public schools in Chile. Developmental difficulties were revealed, showing that boys displayed significantly more difficulties in their social skills when compared to girls, and that kindergartners displayed significantly more developmental difficulties than first graders. The children’s developmental difficulties in executive functions, social skills, and emotional/behavioral problems exhibited interactions between gender and school age. The findings were discussed in terms of current conceptualizations of both executive functions and self-regulatory processes. These processes and functions are configured early in development, are gradually consolidated over the course of school age, and can be strengthened or weakened by conditions experienced in childhood. Early screening of developmental difficulties from the parents’ perspective would facilitate early detection of problems, as early as in kindergarten, and considering the normal adaptable development of children.

## 1. Introduction

The early detection and diagnosis of developmental difficulties in early childhood has led to better prognoses when they are linked to early interventions [1,2]. The term “developmental difficulties” refers to a range of problematic behaviors experienced by infants and young children. These difficulties can be described as delays in language, social-affective, cognitive, behavioral, and motor development [3].

In response, a growing number of instruments have been developed to detect developmental difficulties, such as developmental delays, learning difficulties, and social adjustment problems [4,5]. However, these instruments have also characterized difficulties in typically developed children, where such difficulties may not necessarily represent a developmental disability. Thus, particular patterns of problematic behavior can be detected with these instruments [6]. However, many of them detect one or a few areas of development [5,7]; whereas, more comprehensive screening assessment instruments are still lacking. These comprehensive instruments have become relevant because they facilitate appropriate professional referral, early treatment, better prognosis, and efficiency of support services [8]. Therefore, there is a need for instruments able to capture problematic behaviours that do not necessarily constitute a developmental disability.

Screening instruments can report early signs of developmental problems. These instruments can be easily completed by parents or teachers, in contrast to more time- and resource-consuming tools such as direct assessments and observations by trained practitioners [9]. Parents, in particular, have been considered reliable and valid sources of information regarding their child’s development [7,10,11]. Parents’ reports have been shown to be comparable to laboratory-based measures [7], highly sensitive and specific in detecting child problems [10], and particularly sensitive when reporting global developmental delays [11]. Thus, parents’ reports can contribute to the early detection of potential problems for further evaluation by developmental professionals.

### 1.1. The Five to Fifteen (FTF) Questionnaire

The FTF questionnaire is one of the screening instruments to use parents’ reports to aid in the screening of developmental problems, not only with clinical samples, but also with characterizing specific behavioural patterns in typically developing children [4]. Given that parents complete questions on several areas of their children’s development, the FTF provides a description of the child’s behavioural and neurocognitive status, including both strengths and weaknesses easily observable by parents on a daily basis [12,13]. Thus, it gives the clinician a quantitative and qualitative characterization of the child’s development in eight different areas, i.e., motor skills, executive functions, perception, memory, language, learning, social skills, and emotional/behavioral problems [4,12].

The FTF questionnaire was developed by the Nordic consensus group in the early-mid 1990s [4]. From its creation, this questionnaire has shown appropriate psychometric properties in Nordic clinical [13,14] and non-clinical samples [4,15]. Given that this questionnaire has shown to be a useful screening tool for including a wide range of developmental areas covering the age range from 5 to 15 years, it has also been translated into Spanish for evaluation in other countries, such as Spain [16] and Chile [17].

This questionnaire has been demonstrated to be a reliable and valid instrument. The reliability has been measured by internal consistency and temporal stability. Regarding internal consistency, Cronbach’s alpha coefficient has been widely used (Table 1). This coefficient for the general scale has been excellent [14,16,17]. The internal consistency of the FTF domains has also been excellent when evaluating samples of typically developed children [15,17] and clinical samples of children with attention deficit hyperactivity disorder (ADHD) and other neuropsychiatric conditions [13]. In terms of the temporal stability of the FTF questionnaire, Kadesjö, Janols [4] computed test-retest reliability with six to eight weeks between the first and the second application. The Pearson coefficient for the domains ranged from 0.74 (memory) to 0.91 (executive functions). This study also showed a good level of agreement between the information provided by fathers and mothers (Pearson correlations ranging from 0.67 to 0.85 for domains).

In terms of criterion-related validity, Trillingsgaard, Damm [13] compared FTF domains with some indexes from the Wechsler Intelligence Scale for Children–third version (WISC-III) [18]. FTF domains were significantly correlated to WISC-III indexes (language, learning, and perception), but executive functions had no significant correlation with the comparable index. To test concurrent validity, Bohlin and Janols [12] compared the FTF questionnaire with the child behaviour checklist (CBCL) developed by Achenbach [19]. In those domains covered by both instruments, they found statistically significant correlations, oscillating from 0.20 to 0.81.

Korkman, Jaakkola [6] estimated predictive validity by using the neuropsychological assessment scale (NEPSY) developed by Korkman, Kirk [20], in a sample of children at neuropsychological risk. Significant correlations were obtained for the domains fine motor skills, attention and impulsivity, perception, memory, and language. Consistently, children with higher FTF scores (i.e., the group of children at risk) exhibited lower NEPSY scores, a characteristic pattern of neuropsychological disorder. The FTF questionnaire showed good sensitivity (93%); however, specificity was not good (35%). Thus, this questionnaire showed a higher capacity to detect children that certainly have deficits in some areas of development; however, the capacity to detect children that do not have deficits was lower [6].

Bohlin and Janols [12] analyzed the construct validity using exploratory factor analysis (EFA). When the matrix was submitted to an oblique rotation, two factors emerged. In a clinical sample (children diagnosed with ADHD), Bruce, Thernlund [21] conducted a principal component analysis with varimax rotation and six factors emerged, explaining 73% of the total variability. A similar six-factor structure was supported by Lambek and Trillingsgaard [22] based on a confirmatory factor analysis (CFA) that they conducted in a Danish population sample, while a five-factor structure emerged from a previous EFA. Finally, Beltrán-Ortiz, Todd De Barra [17] conducted the same EFA that was implemented by Bohlin and Janols [12], but using a Chilean sample. Rather than two factors, four factors were extracted from this new analysis, explaining 65.43% of the variance. All these factors were highly correlated to each other, a phenomenon indicative of a shared commonality. Thus, despite the cultural differences between the Nordic and the Spanish-speaking countries [23], the Spanish version of the FTF questionnaire replicated the Nordic psychometric properties and confirmed that this questionnaire was also suitable to these contexts [17]. Table 2 provides a summary of the outcomes of studies evaluating the construct validity of the FTF questionnaire.

In terms of the capacity of this instrument to detect developmental difficulties by gender and age, several studies have reported that the FTF questionnaire behaves differently between groups [4,12]. In terms of gender differences in non-clinical samples, studies have reported that boys have greater difficulties compared to girls in all eight domains [12,15]. Other studies reported boys having greater difficulties in only two domains: executive functions and motor skills [4], and two sub-domains: hyperactivity-impulsivity and expressive language, compared to girls [24]. In contrast, in clinical samples, girls have shown significantly greater difficulties than boys in the language and social skills domains [13]. Another study reported that boys showed more difficulties than girls in motor skills and perception, whereas girls showed more difficulties than boys in the learning domain and reading and writing subdomain [21]. Thus, boys have shown consistently to have greater difficulties than girls in non-clinical samples; whereas this trend is less clear in clinical samples.

Regarding the discriminant capacity by age, the FTF questionnaire has shown that as children get older, their scores in the FTF begin to decrease [4,12], reflecting an adaptive development [17]. For instance, 8-year old children showed fewer difficulties than 6- and 7-year old children combined, particularly in perception, memory, and language [15]. This decline in developmental difficulties has been also reported in other studies including a wider age range, i.e., Kadesjö, Janols [4] and Beltrán-Ortiz, Todd De Barra [17]. Thus, it seems that the FTF questionnaire has been able to detect the gradual reduction of developmental difficulties by age, independently of the age range used to compare groups.

Despite the evidence supporting age and gender differences, only a few studies have reported the effects of their interaction [12,17]. Bohlin and Janols [12] only reported a significant interaction for the Reading and Writing subdomain, showing that boys had increasing problems with age, whereas girls had fewer problems with age. Beltrán-Ortiz, Todd De Barra [17] showed that girls had a superior performance in executive functions than boys in most of the grades. Even when boys performed better than girls in some grades, it was obvious that the age and the gender were interacting. This interaction may be attributed to the way in which the sample was selected and its small size, the variability of the school environment, or cognitive mechanisms underlying the children’s adaptive functioning. Considering that the sample size and the selection process of participants could explain the lack of similarity in the interactions between the studies, the evidence of interaction would be less questionable. On the other hand, attributing this interaction to environmental factors, when schools were quite homogenized in their structural characteristics and pedagogical resources, may not be the most adequate explicative factor. Therefore, it is possible that certain cognitive mechanisms underlying the adaptation process are interacting differently in school settings, depending on the gender of the children [25].

### 1.2. The Transition from Preschool to Primary School

The transition from preschool to primary school is a sensitive period for early childhood and education research [26,27]. A successful transition may translate into positive socio-emotional development, social skills, and positive school trajectories [26,28], whereas an ineffective transition may lead to school failure and social adjustment issues [29]. Children themselves have perceived this transition as both exciting, with the new learning environment, and worrying, with concerns about the unknown [30,31]. Gender also seemed to play a role during the transition, as girls have shown better adjustment to school than boys [32,33].

In Chile, like other countries, kindergarten is the last level of preschool education, and first grade is the first level of primary education. Children in their preschool education access play-based teaching strategies, stimulating environments, continuous interactions with peers, and generally more than one educator per class. In contrast, when children transition to first grade they spend more time in rigid environments, sitting at their desks, and copying in their notebooks the content delivered by one teacher, and interacting less time with peers [34]. Although preschool education introduces general development, social skills, and learning of basic knowledge in preparation for primary education [35], there is no continuity between the kindergarten and first grade curriculum, and this lack of alignment negatively affects children’s adaptation [29,34].

Thus, if the transition to primary school represents such a risk for child development, the use of screening instruments would facilitate early detection of such issues to prevent further risks and maximize the strengths [2,32]. The FTF questionnaire, in particular, would facilitate the early monitoring of normal and problematic developmental difficulties, such as motor, cognitive, communicative, social, and emotional/behavioral areas, during this transition.

## 2. The Current Study

Thus, we decided to further analyze how developmental difficulties presented in different school ages (kindergartners vs. first graders) interact with gender (boys vs. girls), and how this effect was depicted by the FTF questionnaire’s domains in a non-clinical sample. In these terms, this study had an incremental character, because it aimed to characterize the interaction between gender (boys and girls) and school age (kindergartners vs. first graders). To our knowledge, there was no evidence with the original FTF questionnaire and the Spanish version about how gender and school age interact during this transition, and this study aimed to fill this gap. Based on previous studies of non-clinical samples [4,12,15], we predicted that boys would show greater developmental difficulties compared to girls. In line with previous findings [4,12], we also expected that children from kindergarten would display greater difficulties than children from first grade. Given that previous studies have reported interactions by gender and age [12,17], we expected to identify interactions by gender and school age; in which self-regulatory processes and executive function are the main constituent.

The goal of this study was to demonstrate that certain developmental difficulties were expressed differently for boys and girls, while they moved from kindergarten to the first year of primary school. A possible explanation for this interaction was due to the way in which school environments are organized. However, this alternative was discarded because the primary school affects boys and girls in a homogeneous way. The other more plausible explanation, related to neurodevelopmental antecedents, would be that temperamental traits associated with self-regulatory functioning were installed earlier, and in a more stable way, in girls, making them express fewer difficulties when they transition from kindergarten to the first year.

## 3. Methods

### Participants

Participants were 541 parents of children attending kindergarten and first grade in public schools in Chile. The families were recruited from urban areas of the cities of Santiago and Talca. The exclusion criteria were: children with auditory, visual, and developmental deficits; below-average academic achievement; and behavioral problems, according to the teacher’s opinion. Thus, in this sample 264 (48.8%) were parents of girls and 277 (51.2%) of boys. While 197 parents (36.41%) had their children in kindergarten (53.3% parents of girls and 46.7% parents of boys), 344 parents (63.59%) had their children in first grade (46.2% parents of girls and 53.8% parents of boys). The mean age was 5.57 (*SD* = 0.54) for kindergarten and 6.44 (*SD* = 0.54) for first grade children.

## 4. Procedure

The recruitment involved meetings with primary school principals to obtain their consent to conduct the research in their schools, which followed individual meetings with educators/teachers of kindergarten and first grade to explain the research project and organize contact with parents to complete the questionnaires. Parents were contacted during parent’s meetings at schools, where they completed questionnaires using paper and pencil, after which they provided their written consent. The completion time varied between 30 and 45 min, and parents were required to complete the questionnaire in one sitting. The study received ethical clearance from the Scientific Ethic Committee of the Universidad de Talca, according to IRB # 1161533.

## 5. Measure

The five to fifteen (FTF) questionnaire. This questionnaire (181 items) measures eight domains of a child’s behavioural and neurocognitive development from their parent’s report, i.e., motor skills, executive functions, perception, memory, language, learning, social skills, and emotional/behavioral problems. Each item uses a three-point Likert scale (0 = does not apply; 1 = applies sometimes or to a certain extent; 2 = definitively applies). The total score of each domain was calculated by the mean score of these items. Higher scores indicated higher levels of dysfunction in that domain [4,17]. A list of all 181 items of this questionnaire can be obtained from Kadesjö, Janols [4] for the English version and Beltrán-Ortiz, Todd De Barra [17] for the Spanish version.

Overall, the Spanish version of the FTF questionnaire in our study showed a very good internal consistency (Cronbach’s alpha coefficient) for the general scale and the eight domains for kindergarten and first grade groups (see Table 3).

Using an exploratory factor analysis (EFA) with Oblimin rotation and unweighted least squares as an extraction procedure, two factors emerged: *KMO* = 0.90; *χ*^2^ = 2277.76; *df* = 28; *p* < 0.001. Before rotation, the first and second factors explained 52.87% and 12.50% of the variance, respectively. Based on the structure matrix (see Table 4), the first factor was labelled as general developmental difficulties, while the second one was labelled as language difficulties. Both factors were uncorrelated (*r* = −0.02; *p* = 0.59). The estimated standardized scores of factors 1 and 2 were saved for each participant.

### Statistical Analyses

Data analyses were conducted using Windows IBM SPSS 21. A factorial ANOVA was conducted to specify the magnitude of each main effect and the effect of the interaction between children’s gender and school age. The second analysis aimed to evaluate the discriminant capacity of the FTF domains to differentiate the four groups.

## 6. Results

Means scores and standard deviations of the domains and subdomains of the FTF questionnaire separated by gender and school age are presented in Table 5.

A two-by-two ANOVA was conducted using gender and school age as between-subject factors. An interaction effect was found between gender and school age for executive functions, *F* (1537) = 4.04, *p* < 0.05, *η*_p_^2^ = 0.01; social skills, *F* (1537) = 4.89, *p* < 0.05, *η*_p_^2^ = 0.01; and emotional/behavioral problems, *F* (1537) = 9.52, *p* < 0.01, *η*_p_^2^ = 0.02. For executive functions and social skills, both girls and boys from kindergarten had a similar average score; however, first grade girls decreased their average score (they showed more executive functions), and first grade boys kept the same performance as kindergarten boys. For emotional/behavioral problems, kindergarten boys showed lower scores than kindergarten girls. However, a different trend was observed with first grades students, boys had higher scores than girls.

In terms of gender differences, a main effect was found for social skills, *F* (1537) = 4.71; *p* < 0.05, *η*_p_^2^ = 0.01, in which boys had a lower performance than girls. Regarding school age, a main effect was reported for motor skills, *F* (1537) = 17.53, *p* < 0.001, *η*_p_^2^ = 0.03; perception, *F* (1537) = 15.95, *p* < 0.05, *η*_p_^2^ = 0.01; memory, *F* (1537) = 13.74, *p* < 0.001, *η*_p_^2^ = 0.02; language, *F* (1537) = 25.25, *p* < 0.001, *η*_p_^2^ = 0.05; learning, *F* (1537) = 20.08, *p* < 0.001, *η*_p_^2^ = 0.04; and social skills, *F* (1537) = 4.08, *p* < 0.05, *η*_p_^2^ = 0.01. In all cases, kindergarten children had a lower performance than first grade children.

The same ANOVA was implemented with the standardized scores of factors 1 and 2 that previously were saved for each participant. In terms of the general factor of developmental difficulties (factor 1), there was an interaction effect, *F* (1537) = 6.13; *p* = 0.014, *η*_p_^2^ = 0.01. In kindergarten, girls (M = 0.22) and boys (M = 0.14) did not show any difference, *F* (1537) = 0.37; *p* = 0.54. However, first grade girls showed a negative score (M = −0.28) compared with first grade boys (M = 0.05), indicating less developmental difficulties. Regardless school-age, first grade children (M = 0.18) had a lower average score in this factor than kindergarten children (M =−0.12), *F* (1537) = 11.89; *p* = 0.001, *η*_p_^2^ = 0.02. Finally, no differences between boys (M = 0.09) and girls (M = −0.03) were found, *F* (1537) = 2.26; *p* = 0.13.

In term of language difficulties (Factor 2), kindergarten children expressed (M = 0.28) more difficulties than first grade children (M = −1.61), *F* (1537) = 24.96; *p* = 0.001, *η*_p_^2^ = 0.04. No differences between boys (M = 0.05) and girls (M = 0.07) were found, *F* (1537) = 0.086; *p* = 0.77, nor interactions between gender and school age, *F* (1537) = 0.33; *p* = 0.57.

Overall, the results showed that the children’s performance in three domains (i.e., executive functions, social skills, and emotional/behavioral problems) mostly depended on the interaction of their gender and school age. Second, the factor named general developmental difficulties reproduced this kind of interaction. Third, the differences pointed to gender as the relevant variable only for social skills. Finally, school age by itself was responsible for differences in six domains and the factor labelled language difficulties.

## 7. Discussion

This study aimed to compare the developmental difficulties displayed by boys and girls, and kindergartners and first graders, and the interaction of those difficulties by gender and school age. Although previous studies have reported developmental difficulties influenced by gender [12,24] and age [17], the capacity of the FTF questionnaire to capture them during the transition to primary school remained unclear. Overall, the Spanish version of the FTF questionnaire was able to detect some developmental difficulties when comparing groups by gender, school age, and the interaction of both variables.

In terms of developmental difficulties per gender, findings only showed that boys were socially less skilled than girls, which was consistent with previous studies [4,12,15]. This finding only partially supported our hypothesis, as we expected a wider range of differences, as shown in the literature [12,15]. Still, there may be some underlying elements behind the difficulties in social skills. It has been reported that girls have shown higher empathy [36], more cooperation with others [32], and more interactions with peers [37] than boys. Thus, it may be that other communication skills may leave girls in a better position compared to boys at early ages.

Regarding school age, we confirmed that kindergarten children showed lower performance than children from the first grade in six of the eight domains covered by the FTF questionnaire (i.e., motor skills, perception, memory, language, learning, and social skills). This outcome was coherent with previous findings showing an adaptive improvement in some developmental factors [4,12]. For example, when children perceived that they were physically competent, they tended to maintain attempts to prove this competence, having a positive impact in their motor skills [38]. Thus, it is possible that as long as children keep trying to master their motor skills and other skills, they will be able to show improvements over time.

Several interactions were reported for age and school age. In line with another study [17], this study also found that there are differences in developmental difficulties in executive functions depending on gender and school age. Although all kindergarten children presented similar executive skills, girls showed a better capacity for complex cognitive processes compared to boys by the time they were in first grade. A similar trend applied to social skills. However, the interaction for emotional/behavioral problems had a different trend. If girls in kindergarten had a worse emotional and behavioural condition compared to boys, it became better by first grade. By putting them together, it seemed that preschool girls possessed cognitive and social skills comparable to boys, but these skills may translate into more cognitive and social demands, leading to increased emotional and behavioural problems. However, these problems were lower by the time they entered primary education, in comparison to first grade boys. The higher level of emotional and behavioural issues presented in first grade boys may be due to their greater difficulties in adjusting to the transition to school [32,33]. These difficulties can be attributed to a lack of executive functions and self-regulatory mechanisms.

Current conceptualizations of executive functions have been partially incorporated by the FTF questionnaire. For example, executive functions include top down mechanisms of inhibitory and interference control, working memory, and cognitive flexibility [39,40]. Although the FTF questionnaire includes items measuring aspects related to attention and inhibitory control, it does not explicitly take into account aspects of working memory and cognitive flexibility. Thus, the interaction observed between gender and school age during the transition from kindergarten to first grade was mainly explained by items related to one of the three mechanisms that are usually used to describe executive functions in the literature. In this way, both girls and boys had a similar average score in kindergarten; however, girls decreased their average score in comparison to boys during first grade. This outcome may be due to the fact that they may express more of an inhibitory and interference control rather than more working memory and cognitive flexibility (Diamond, 2013).

A similar effect occurred with social skills. In the FTF questionnaire, social skills are measured through a constellation of behaviors and symptoms that affect the communicative area. Most of are behaviors learned and trained in social interaction with others [41]. Even when they overlap with communicative behavior, some aspects are more strongly related to self-regulatory behavior [40]. In that case, first grade girls decreased their average score in social skills (indicative of less difficulties); because they expressed more planning behaviors than boys.

Our contribution was to analyze in detail the developmental difficulties given by differences per gender and school age and their interaction. To our knowledge, this was the first study to examine the capacity of the FTF questionnaire to differentiate developmental areas by gender and school age during the critical period of transition to primary school. We also used a community sample, which may better represent typically developed children. Our study used all eight FTF domains to compare groups, and these domains were shown to be strongly reliable factors. Thus, the group comparisons provided a comprehensive approach by including a wide range of developmental areas, ranging from cognitive to social skills.

Our study also had some limitations, which may lead to future research. We used a cross-sectional design by comparing two groups of children of different school ages. Thus, future studies may follow a cohort of children during their transition to primary school and beyond using screening measures such as the FTF questionnaire, due to the benefits of using a longitudinal design [42]. This study used the Spanish version in a Chilean sample. Developmental difficulties during the transition to primary school should also be explored by using the versions of the FTF questionnaire in other languages, countries, and cultural contexts to contribute to the broader child development literature. Although parent rating scales have shown to be reliable and valid [7,11] and comparable to teacher ratings (Lambek and Trillingsgaard, 2015), this was our only source of information. Thus, future research should also incorporate other methods, such as professional assessments and observations [9]. Our study only included parents’ reports of typically developed children. Thus, future studies may need to explore the extent to which our findings would be applicable to clinical samples, i.e., children with developmental disorders or mental disorders. Given that previous studies have reported interactions in some developmental areas during the transition to school [43] and later on [44], future studies may also explore the interaction between the FTF questionnaire domains. Our study only focused on children’ development during the transition to primary school. However, there is a body of research highlighting other factors playing a role in a successful transition, such as school community, classrooms, relationships, and routine [26,27,45]. Thus, future research can explore a wider range of factors involved in this transition, in addition to developmental screening instruments.

There are several implications to consider based on our findings. The FTF questionnaire has shown to be a screening tool able to identify early developmental difficulties in children’s transition to primary school. Given that this questionnaire has discriminated between clinical and non-clinical samples [16,24], it has also been suitable to monitor the normal progress of child development in a wide range of development areas [17,21,22] and from kindergarten until entering adolescence [17]. Based on the developmental difficulties reported in this study and its differences by gender, school age, and their interaction, more efforts are needed to provide personalized support to girls and boys in their transition to primary school. These efforts may imply designing, implementing, and adapting existing interventions addressed to children [46,47], teachers [48,49], and parents [50,51]. For instance, Walk, Evers [48] reported a training program for teachers focused on executive functions, showing significant gains in their preschooler students’ executive functions. Thus, if preventive actions involve not only children but also those adults around them, this better support may mitigate developmental challenges when transitioning to primary school by encouraging a stronger family–school partnership [26,52].

In conclusion, the FTF questionnaire was able to identify developmental difficulties during children’s transition to primary school. Boys showed worse social skills compared to girls. Kindergarten children had a lower performance than first grade children in several domains. Executive functions, social skills, and emotional/behavioral problems were significantly influenced by the interaction of gender and school age. The FTF questionnaire seemed to be a tool suitable for the early detection of developmental issues and for monitoring differences during periods as critical and brief as the transition to primary school. This instrument might provide teachers and professionals with an accessible, comprehensive, efficient, and economical tool for the early detection and prevention of possible developmental difficulties at school, while engaging parents’ perspective as reliable informants of their children’s development.

## Figures and Tables

**Table 1 ijerph-18-03958-t001:** Reliability by Internal Consistency of the Five-to-Fifteen (FTF) Questionnaire from Previous Studies.

		Internal Consistency (Cronbach’s Alpha Coefficient)	
Studies	Country	General Scale	Domains
Kadesjö et al. (2004) ^a^	Sweden		0.86–0.96
Airaksinen et al. (2004) ^a^	Finland		0.84–0.99
Trillingsgaard et al. (2004) ^b^	Denmark		0.84–0.93
Rodríguez et al. (2010) ^c^	Spain	0.98	
Beltrán-Ortiz et al. (2012) ^a^	Chile	0.98	0.83–0.93
Illum and Gradel (2014) ^b^	Denmark	0.96	
Lambek and Trillingsgaard (2015) ^a^	Denmark		0.85–0.96

Note. ^a^ Sample of typically developed children. ^b^ Sample of children with ADHD, neuropsychiatric disorder, and/or other developmental disorder/disability (clinical sample). ^c^ Sample included both typically developed children and a clinical sample.

**Table 2 ijerph-18-03958-t002:** Construct Validity of the FTF Questionnaire from European and Chilean Studies.

	Construct Validity	
Studies	No. Factors	Factors
Bohlin and Janols (2004) ^a^	2	Learning difficulties
		Socio-emotional problems
Bruce et al. (2006) ^b^	6	Cognitive skills
		Motor/perception
		Emotion/socialization/behaviour
		Attention
		Literacy skills
		Activity control
Beltrán-Ortiz et al. (2012) ^a^	4	General development
		Socioemotional and control problems
		Cognitive-motor and language development
		Communication and academic learning
Lambek and Trillingsgaard (2015) ^a^	6	Attention
		Cognitive skills
		Motor/perception
		Emotion/socialization/behaviour
		Activity control
		Literacy skills

Note. ^a^ Sample of typically developed children. ^b^ Sample of children with ADHD, neuropsychiatric disorder and/or other developmental disorder/disability (clinical sample).

**Table 3 ijerph-18-03958-t003:** Reliability by Domains of the FTF Questionnaire in the Current Study.

Domains (Number of Items)	Kindergarten (*n* = 197)	First Grade (*n* = 344)
Motor skills (17)	0.83	0.81
Executive functions (25)	0.91	0.90
Perception (18)	0.77	0.80
Memory (11)	0.79	0.84
Language (21)	0.91	0.91
Learning (29) *	0.81	0.93
Social skills (27)	0.90	0.91
Emotional/behavioural problems (33)	0.92	0.92
Total	0.98	0.98

Note. * The learning domain, reading-writing (8 items), and math (5 items) subdomains were not applicable for kindergarten children, 16 items were included for this group for this domain.

**Table 4 ijerph-18-03958-t004:** FTF Questionnaire Structure Matrix of Factorial Saturations.

Domains/Factors	F1	F2
Social Skills	0.84	
Executive functions	0.81	
Perception	0.80	
Learning	0.79	
Emotional/behavioural problems	0.76	
Memory	0.74	
Motor skills	0.69	
Language		1.00
Eigenvalue	4.23	1.00
% Explained variance before rotation	52.87	12.50

Note. Both factors are uncorrelated.

**Table 5 ijerph-18-03958-t005:** Means and Standard Deviations (SD) for Domains, by School Age and Gender.

FTF Domains	Boys		Girls		School Age	Gender	School Age * Gender
	Kinder (*n* = 92)	First Grade (*n* = 185)	Kinder (*n* = 105)	First Grade (*n* = 159)	*F*	*F*	*F*
Motor skills	0.37 (0.28)	0.31 (0.27)	0.39 (0.35)	0.24 (0.23)	17.53 **	1.02	2.82
Executive functions	0.65 (0.36)	0.67 (0.44)	0.68 (0.40)	0.55 (0.38)	2.55	2.06	4.04*
Perception	0.50 (0.27)	0.44 (0.28)	0.53 (0.31)	0.38 (0.28)	15.95 **	0.61	3.62
Memory	0.47 (0.34)	0.41 (0.38)	0.47 (0.37)	0.30 (0.28)	13.74 **	2.91	3.31
Language	0.45 (0.37)	0.32 (0.32)	0.48 (0.37)	0.31 (0.30)	25.25 **	0.08	0.31
Learning	0.65 (0.34)	0.55 (0.39)	0.67 (0.41)	0.47 (0.31)	20.08 **	0.63	2.65
Social Skills	0.32 (0.32)	0.33 (0.31)	0.33 (0.30)	0.21 (0.23)	4.08 *	4.71 *	4.89 *
Emotional/behavioural problems	0.27 (0.28)	0.35 (0.30)	0.35 (0.30)	0.27 (0.27)	0.00	0.00	9.52 **

Note. * *p* < 0.05, ** *p* < 0.001.

## Data Availability

The data that support the findings of this study are available from corresponding author, upon reasonable request.

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
