# Peer review of "Screening of Developmental Difficulties during the Transition to Primary School"

_ijerph, 2021, doi:10.3390/ijerph18083958_

Round 1

Reviewer 1 Report

This is an interesting and robust evaluation of a psychometric instrument with valid and reliable properties. It is obviously a useful diagnostic tool for parents, teachers and support staff in addressing young children's educational transitional and wider developmental problems. The paper has merit in terms of good conventional structure with a clear rationale / problem, an appropriate, design and analyses, critical discussion / conclusion. Language used is mostly clear and succinct (some minor translation issues). It should be made clear in the abstract and throughout that parents perceptions are being evaluated. There is significant validity material in the introduction / background material which although useful, maybe too elaborate in relation to the overarching objective of the paper. The main interest is in the interactions between the age and gender of a large sample group of children, understanding differences in subdomains and what can be done to address these issues. 

Reviewer 2 Report

The present manuscript, entitled “Screening of developmental difficulties during the transition to primary school”, focuses on early screening of developmental difficulties in children form parents’ perspectives. Using the Five-to-Fifteen (FTF), developmental difficulties were revealed, showing interactions between gender and school age regarding difficulties in Executive functioning, Social skills and Emotional/behavioural problems. As such, this study is very interesting not only for research in the field of child development, but also for clinical practice of professionals who work with children, in particular for preventive child health care.

The Five-to-Fifteen questionnaire has been in use for more than a decade in Scandinavian countries for both clinical and research applications and has shown fairly good validity and reliability. For screening purposes, the FTF has also a good sensitivity, but specificity appeared to be rather low. However, especially in preventive child health care the FTF can be extended with developmental assessment and/or longitudinal repeated measures (monitoring), assuming that specificity and sensitivity will improve. In this respect, it is somewhat surprising that the authors opted for a cross-sectional design. After all, children attending kindergarten will be first graders one year later, so an interval of only one year could provide a longitudinal design respectively the opportunity of repeated measures.

Regarding the chosen design, it is a missed opportunity that parents were the only informants, while the perspectives of teachers of kindergarten and first grade were not involved in this research. Yet, the recruitment of participants was achieved through schools, so involvement of teachers would have been quite obvious.

Specific remarks:

Line 287: “The families were recruited from urban 287 areas of Santiago and Talca cities”. The manuscript does not tell us how many participants were recruited from Santiago respectively from Talca. Moreover, information about the demographical differences between these cities is lacking. For instance, population density is 9 to 10-fold higher in Santiago versus Talca, if I am not mistaken.

Line 290: “The children’s Socio-Economic Status (SES) was categorized as low and medium”. Given the research question, the SES is an important variable. However, SES is not included in the analyses as presented. In the discussion, SES is lacking too. Both aspects, demographical characteristics and SES, are important environmental aspects and these aspects should be addressed within the context of possible confounding factors. How was the geographical distribution of the participants? More in detail, how was de SES distribution in kindergarten resp. schools? For instance, if the children in kindergarten were predominantly children from socially disadvantaged backgrounds and first graders from medium SES, this could have affected the outcome variables. The authors should provide a clear description of the population from which the study participants were selected or recruited, including demographics, location, and time period.

Line 291: The applied exclusion criteria are all described as child characteristics. However, given the fact that the FTF is a parent reported tool, this raises the question whether also parent related exclusion criteria should have been identified (like health illiteracy parents, language difficulties, cognitive limitations of parents). Authors cannot rule out that some parents may have had difficulties due to their own cognitive problems. As such, this would be another limitation.

Line 307: “The study received ethical clearance from the Ethics Committee of the University”. Registration number and date of the ethical decision should be added. Furthermore, presumably the University of Talca is meant, and not the University of Queensland (affiliation of the first author). Was it clear for parents that participation was on the voluntary basis? Did parents fill in a written informed consent?

List of references is at an appropriate level.

Round 2

Reviewer 2 Report

As a response to the revised version, most of my comments are addressed in an acceptable way. However, my question regarding SES remains partly unanswered. Regarding participants (lines 287-290), it is stated that “the children’s Socio-Economic Status (SES) was categorized as low and medium. In the discussion (lines 505-509) the revised text tells us now: “Similarly, children attended public schools, which implies similarities in terms of some demographic and educational variables, e.g., SES and curriculum. However, this study did not explore the impact of these and other contextual factors […]”. As such, similarities in terms of some demographic and educational variables are assumptions not supported by evidence. Moreover, the question remains open why a SES -variable was categorized without further evaluation in the context of the present study. It does not make much sense to measure respectively to categorize this variable and, subsequently, to declare that this study did not explore its impact. So, my recommendation is to include this variable in the analyses. Otherwise, remove the text in lines 287-290.
